# The changing epidemiology of human monkeypox—A potential threat? A systematic review

**Eveline M. Bunge**[1], **Bernard Hoet**[2]*, **Liddy Chen**[3], **Florian Lienert**[2], **Heinz Weidenthaler**[4], **Lorraine R. Baer**[5], **Robert Steffen**[6,7]

**1** Pallas Health Research and Consultancy, Rotterdam, The Netherlands, **2** Bavarian Nordic AG, Zug, Switzerland, **3** Bavarian Nordic, Inc., Morrisville, North Carolina, United States of America, **4** Bavarian Nordic GmbH, Martinsried, Germany, **5** Baer PharMed Consulting, Ltd., Skokie, Illinois, United States of America, **6** Epidemiology, Biostatistics and Prevention Institute, WHO Collaborating Center on Travelers' Health, University of Zurich, Zurich, Switzerland, **7** Department of Epidemiology, Human Genetics and Environmental Sciences, University of Texas School of Public Health, Houston, Texas, United States of America

* beho@bavarian-nordic.com

**Data Availability Statement:** All relevant data are within the manuscript and its Supporting Information files.

## Abstract

Monkeypox, a zoonotic disease caused by an orthopoxvirus, results in a smallpox-like disease in humans. Since monkeypox in humans was initially diagnosed in 1970 in the Democratic Republic of the Congo (DRC), it has spread to other regions of Africa (primarily West and Central), and cases outside Africa have emerged in recent years. We conducted a systematic review of peer-reviewed and grey literature on how monkeypox epidemiology has evolved, with particular emphasis on the number of confirmed, probable, and/or possible cases, age at presentation, mortality, and geographical spread. The review is registered with PROSPERO (CRD42020208269). We identified 48 peer-reviewed articles and 18 grey literature sources for data extraction. The number of human monkeypox cases has been on the rise since the 1970s, with the most dramatic increases occurring in the DRC. The median age at presentation has increased from 4 (1970s) to 21 years (2010–2019). There was an overall case fatality rate of 8.7%, with a significant difference between clades—Central African 10.6% (95% CI: 8.4%– 13.3%) vs. West African 3.6% (95% CI: 1.7%– 6.8%). Since 2003, import- and travel-related spread outside of Africa has occasionally resulted in outbreaks. Interactions/activities with infected animals or individuals are risk behaviors associated with acquiring monkeypox. Our review shows an escalation of monkeypox cases, especially in the highly endemic DRC, a spread to other countries, and a growing median age from young children to young adults. These findings may be related to the cessation of smallpox vaccination, which provided some cross-protection against monkeypox, leading to increased human-to-human transmission. The appearance of outbreaks beyond Africa highlights the global relevance of the disease. Increased surveillance and detection of monkeypox cases are essential tools for understanding the continuously changing epidemiology of this resurging disease.

**Funding:** This study was initiated and funded by Bavarian Nordic. The decision to publish the manuscript was made after the report of the paid systematic literature review by Pallas Health Research and Consultancy. The funder had a role in study design, data collection and analysis, decision to publish, and preparation of the manuscript. Baer PharMed Consulting, Ltd received funding for the development of the manuscript.

**Competing interests:** I have read the journal's policy and the authors of this manuscript have the following competing interests: Bernard Hoet, Liddy Chen, Florian Lienert, and Heinz Weidenthaler are employees of Bavarian Nordic. Eveline M. Bunge is an employee of Pallas Health Research Consultancy. Lorraine R. Baer is an employee of Baer PharMed Consulting, Ltd., which received funding for the preparation of the manuscript. Robert Steffen has been paid for lectures and participation in advisory boards by various vaccine manufacturers, including Bavarian Nordic. Please note that the study was fully funded by Bavarian Nordic with two contracts, first with Pallas Health Research and Consultancy to perform the systematic review, second with Baer PharMed Consulting, Ltd to support manuscript development. It is only after reviewing the data acknowledging the public health importance that the authors decided to submit a manuscript on the subject. The following authors received salary from the funder (BH, LC, FL, HW). The funder had a role in study design, data collection and analysis, decision to publish, and preparation of the manuscript.

## Author summary

Monkeypox, a zoonotic disease caused by an orthopoxvirus, results in a smallpox-like disease in humans. We conducted a systematic review to assess how monkeypox epidemiology has evolved since it was first diagnosed in 1970 in the Democratic Republic of the Congo. In total, human monkeypox has now appeared in 10 African countries and 4 countries elsewhere. Examples include Nigeria, where the disease re-emerged in the last decade after a 40-year hiatus, and the United States, where an outbreak occurred in 2003. The number of cases has increased at a minimum of 10-fold and median age at presentation has evolved from young children (4 years old) in the 1970s to young adults (21 years old) in 2010–2019. This may be related to the cessation of smallpox vaccinations, which provided some cross-protection against monkeypox. The case fatality rate for the Central African clade was 10.6% versus 3.6% for the West African clade. Overall, monkeypox is gradually evolving to become of global relevance. Surveillance and detection programs are essential tools for understanding the continuously changing epidemiology of this resurging disease.

## Introduction

Monkeypox, currently a rare zoonotic disease, is caused by the monkeypox virus, which belongs to the Poxviridae family, Chordopoxvirinae subfamily, and Orthopoxvirus genus [1]. The variola virus (smallpox virus) is closely related [1], and monkeypox disease results in a smallpox-like disease. Historical data have indicated that smallpox vaccination with vaccinia virus (another orthopoxvirus) was approximately 85% protective against monkeypox [2]. However, following the eradication of smallpox in 1980, routine vaccination against smallpox was no longer indicated [3], and it has now been four decades since any orthopoxvirus vaccination program.

The name monkeypox originates from the initial discovery of the virus in monkeys in a Danish laboratory in 1958 [4]. The first case in humans was diagnosed in 1970 in a 9-month-old baby boy in Zaire (now the Democratic Republic of the Congo, DRC) [5]. Since that time, monkeypox has become endemic in the DRC, and has spread to other African countries, mainly in Central and West Africa. Outside of Africa, the first reported monkeypox cases were in 2003 [6] and, at the time of this systematic review, the most recent cases were in 2019 [7,8].

A previous systematic review, which evaluated the literature through summer 2018, described the epidemiology of monkeypox outbreaks [9]. In view of the recent increase in reports from Nigeria and elsewhere, we initiated a new systematic literature review with a focus on the changes in the evolution of the epidemiology of human monkeypox since the first cases in the 1970s through the present day.

## Methods

This systematic review was performed in accordance with international standards for conducting and reporting systematic reviews, including guidelines from the Cochrane Collaboration [10] and the Preferred Reporting Items for Systematic Reviews and Meta-Analyses (PRISMA) [11]. The review is registered with PROSPERO (CRD42020208269).

Searches were performed in MEDLINE (accessed using PubMed), Embase, African Journals Online (AJOL) and the Internet Library sub-Saharan Africa (ilissAfrica), with no language restrictions. All published literature reported through September 7, 2020, the date of last

search, was considered for eligibility. In PubMed, searches included Medical Subject Headings (MeSH) and limits to title and abstract (tiab). The search string used in PubMed was Monkeypox[MeSH] OR "Monkeypox virus"[MeSH] OR monkeypox[tiab] OR "monkey pox"[tiab] OR "variole du singe"[tiab] OR "variole simienne"[tiab] and in Embase was 'monkeypox'/exp OR 'monkeypox virus'/exp OR monkeypox:ti,ab OR "monkey pox":ti,ab. In AJOL and illissAfrica, separate searches were conducted for each of the following terms: monkeypox, variole du singe, and variole simienne. We aimed to explore how monkeypox epidemiology has evolved regarding incidence, case characteristics, clades, transmission, and case fatality rate. We also sought to explore the risk factors for acquiring human monkeypox.

After all articles were identified from the four databases and duplicates removed, screening of the title and abstract was performed in duplicate by two researchers (EMB [author] and BVD). Articles that seemed to contain relevant data for the review objectives, which included all age populations, were selected for full-text screening. Excluded were non-human studies, modelling studies that did not provide original data, articles that focused primarily on smallpox, and articles with data not related to the topics of interest. In cases of doubt, the article was selected for full-text screening. Full text articles were then reviewed to determine whether at least one of the review objectives was met. At this stage, other articles such as conference abstracts or narrative reviews were also excluded. The first 10% of full text articles was critically evaluated in duplicate by two researchers (EMB and BVD), and the remaining 90% were reviewed by EMB. Each article was then further reviewed during data-extraction. Some additional exclusions occurred in this step. For example, for articles with similar results from largely identical data sets, only one article was included, generally the most recent. In some instances, there was a partial overlap of data, such that different articles included the same cases plus some unique cases. In these situations, only the unique cases in each article were included in the data extraction sheet. One researcher (EMB) created the data extraction sheet for the eligible articles, and these were reviewed by a second researcher (RVH). A random check of 10% of the data extraction was performed by RVH.

Articles suitable for extraction from the literature searches were case reports, outbreak investigations, epidemiological studies, and surveillance studies. For these types of articles, no formal checklists for critical appraisal are available, so no formal quality assessments were performed. Information on study quality as reported by the authors of the selected articles were added as comments in the data extraction sheet.

In addition to the four primary search sources, seven sources of grey literature and Google were searched during weeks 41–44 of 2020. These sources were the websites of the World Health Organization (WHO), specifically a review of the Weekly Bulletins on Outbreaks and other Emergencies, United States Centers for Disease Control and Prevention (CDC), Africa CDC, Nigeria CDC, African Field Epidemiology Network, Epicentre, and ProMed. The Google search was performed on the African countries known to have monkeypox cases, including a check on the websites of their ministries of health. No formal search strategy was employed, and therefore no denominator on number of reports is described. One researcher performed the search of the grey literature (LM) and a second researcher (EMB) reviewed the findings and added the relevant information to the data extraction sheet.

## Pooled analysis

For age at monkeypox infection, a weighted average of the median ages per decade was calculated, based on investigations where median age was reported and on single cases where age was presented. In each respective decade, these single cases were treated as a unit, and median age determined.

Data on case fatality rate (CFR) were pooled, and 95% confidence intervals (CIs) were calculated using the binomial (Clopper-Pearson) exact method. Both overall CFR and CFR per clade were calculated. Since specific clade data were not always reported in the literature, we used the geographical spread of the clades as described by WHO [12] to assign the clade variant. Monkeypox cases from the DRC, Gabon, Central African Republic (CAR), South Sudan, and Republic of the Congo were assumed to be of the Central African clade, while cases in all other countries were assumed to be of the West African clade. Cameroon was not included in the number of cases per clade or the calculation of CFR by clade, as WHO has reported that both clades have been detected there [12].

## Case definitions

Case definitions were not standardized across sources, but in general the definitions displayed in Table 1 were used.

## Results

The search strategy yielded a total of 1,995 publications, 129 of which were selected for full-text screening. Of these, 48 articles were suitable for data extraction. An additional 18 records from the grey literature (primarily the WHO website) were also included for data extraction. The PRISMA flowchart of the selection process for the systematic review is in shown in Fig 1.

## Number of reports by country

Monkeypox data from the DRC accounted for approximately one-third of the eligible articles [5,13,14,16–28]. The remaining articles had monkeypox data from the CAR [29–35], United States (US) [36–41], Nigeria [5,42–44], Republic of the Congo [15,45–47], Sierra Leone [42,48,49], Cameroon [5,50], Côte d'Ivoire [51,52], Gabon [53,54], United Kingdom (UK) [55,56], Israel [57], Liberia [42], Singapore [8], and South Sudan [formerly Sudan] [58]. (Note: two articles [5,42] described data for more than one country, therefore the total number of articles per country exceeds 48.) The 18 grey literature reports were from the CAR [59–62], DRC [63–66], Cameroon [65,67,68], Republic of the Congo [69–71], Liberia [72,73], Nigeria [68,74], UK [7], and US [6]. All but two sources [17,45] reported on the epidemiology of monkeypox; these two were peer-reviewed articles on risk factors for acquiring monkeypox.

**Table 1. Monkeypox Case Definitions.**

| Type of case | Definition |
|---|---|
| Suspected | Sudden onset of high fever, followed by a vesicular-pustule eruption presenting predominantly on the face, palms of the hands, and soles of the feet; or the presence of at least 5 smallpox type scabs. |
| Confirmed | Suspected case with laboratory confirmation (Positive IgM Antibody, PCR, or virus isolation). |
| Probable | Suspected case with no possibility of laboratory confirmation, but with epidemiological link to a confirmed case. |
| Possible | Vesicular, pustular or crusted rash, not diagnosed as chickenpox by the family or the health-care provider [13]. History of fever and vesicular or crusty rash [14]. Individual met one of the epidemiologic criteria or demonstrated elevated levels of orthopoxvirus-specific IgM and had unexplained rash and fever and $\geq 2$ other signs or symptoms from the clinical criteria [15]. |

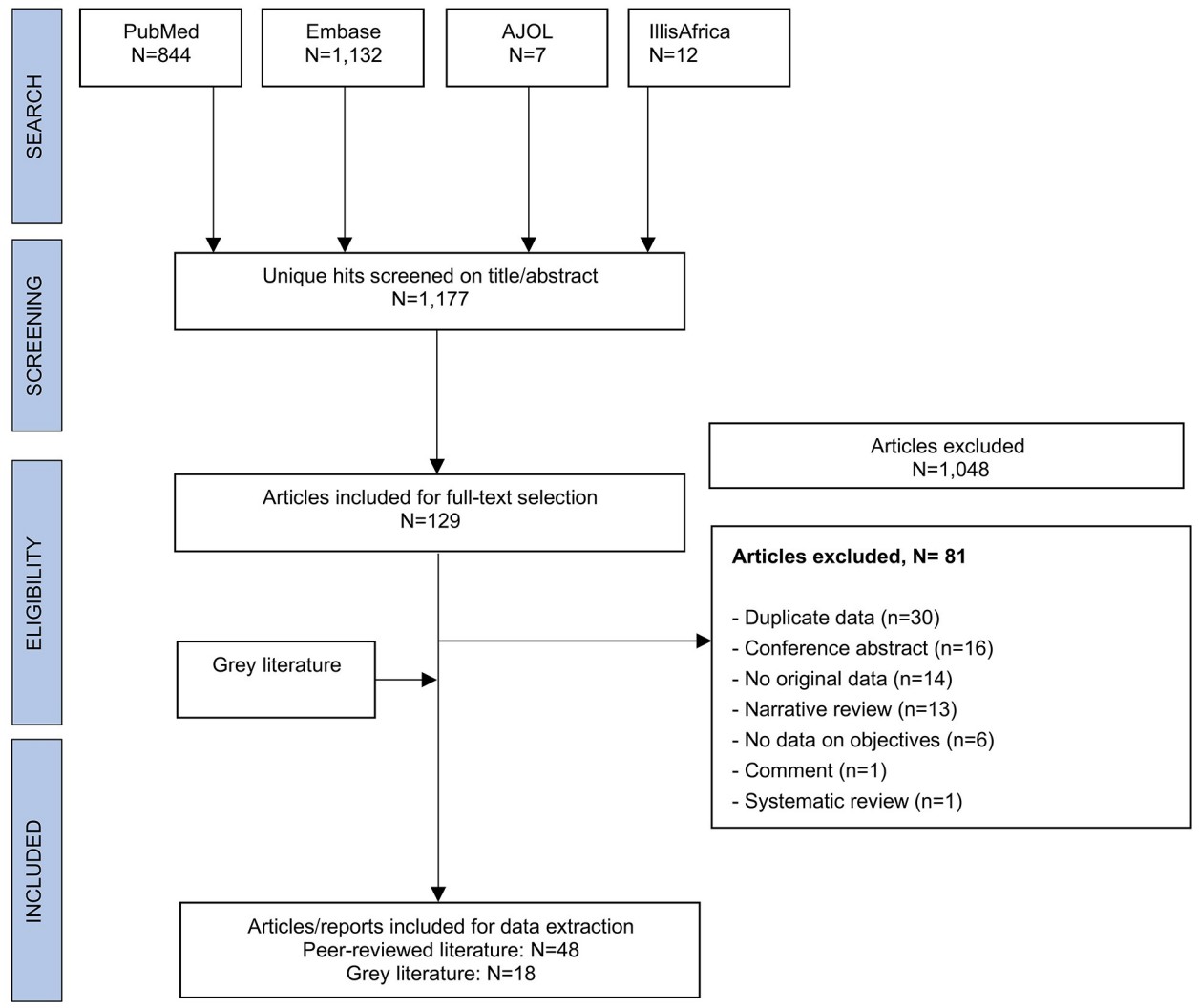

**Fig 1. PRISMA flowchart.**

## Number of cases by country

We identified 28 peer-reviewed articles [5,8,14,15,18,20,21,29–35,42,46–58] and 15 reports from the grey literature [6,7,59–65,67–69,72–74] with data on the number of confirmed, probable, and/or possible monkeypox cases, for a total of 1,347 cases, and an additional 28,815 suspected cases from the DRC. These data are displayed in Figs 2 – 6 (and S1 Table) by decade, starting with the 1970s, when the first cases were detected [5,42,51].

During the 1970s, a total of 48 confirmed and probable monkeypox cases were reported in six African countries, namely the DRC, Cameroon, Côte d'Ivoire, Liberia, Nigeria, and Sierra Leone, with most cases occurring in the DRC (n = 38) (Fig 2).

In the 1980s compared to the 1970s, a 9-fold increase in the number of confirmed and probable monkeypox cases was observed in the DRC (n = 343). In addition, 14 other cases were spread among four other African countries (Fig 3).

Cases continued to increase in the 1990s, with 511 confirmed, probable, and/or possible monkeypox cases reported in DRC and 9 confirmed cases in Gabon (Fig 4).

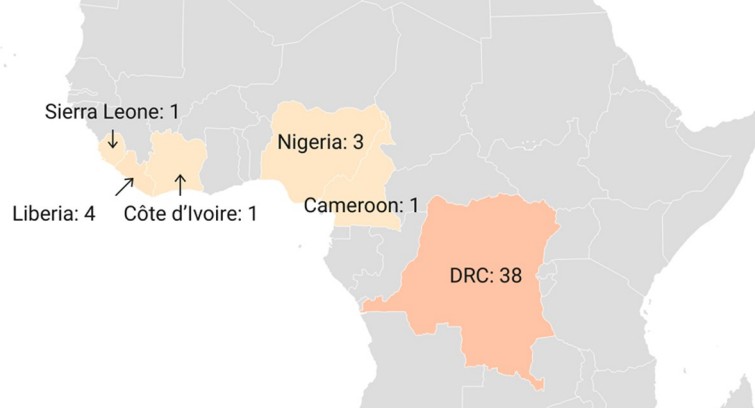

**Fig 2. Number of confirmed, probable, and/or possible monkeypox cases between 1970–1979.** [5,42,51] (base layer of the map: https://datawrapper.dwcdn.net/W7k0L/4/).

Between 2000 and 2009, monkeypox cases were reported in three African countries (DRC, Republic of the Congo, and South Sudan) (Fig 5), but between 2010 and 2019, cases were found in seven African countries (Cameroon, CAR, DRC, Liberia, Nigeria, Sierra Leone, and Republic of the Congo) (Fig 6). Compared to the last three decades of the 20[th] century, outbreaks as of the year 2000 were greater in total number of cases and fewer in singular case reports.

The DRC is the country most affected by monkeypox, and no other country has reported monkeypox cases continuously during the past five decades. As of the year 2000, however, the number of suspected cases, rather than confirmed, probable, and/or possible cases, was primarily reported, as shown in Fig 5 (2000–2009) and Fig 6 (2010–2019). More recently, between January and September 2020, another 4,594 suspected cases were reported for the DRC [66]. The second most affected country is Nigeria, due to the 181 confirmed and probable cases from the outbreak that started in September 2017 [74]. (Note: 183 cases are noted in the Nigeria CDC report [74], but two cases originating from Nigeria were diagnosed in Israel [57] and Singapore [8] and considered travel-related events for these respective countries. The three UK cases that originated in Nigeria [7,55] are not among the 183 cases in the Nigeria CDC report.) The third and fourth most affected countries with confirmed, probable, and/or

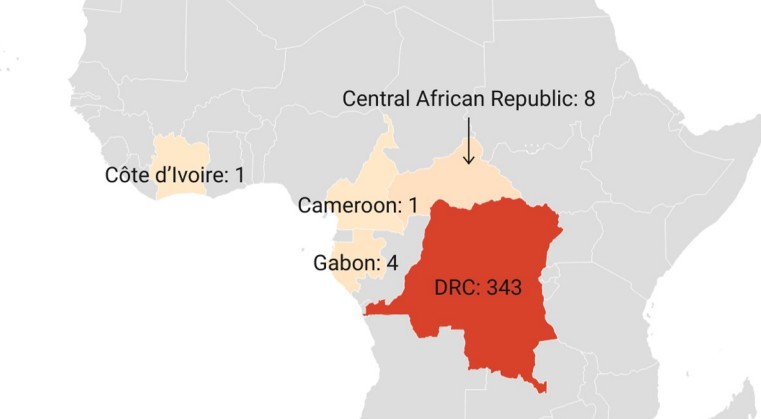

**Fig 3. Number of confirmed, probable, and/or possible monkeypox cases between 1980–1989.** [20,21,31,34,50,52,54] (base layer of the map: https://datawrapper.dwcdn.net/lGHEu/1/).

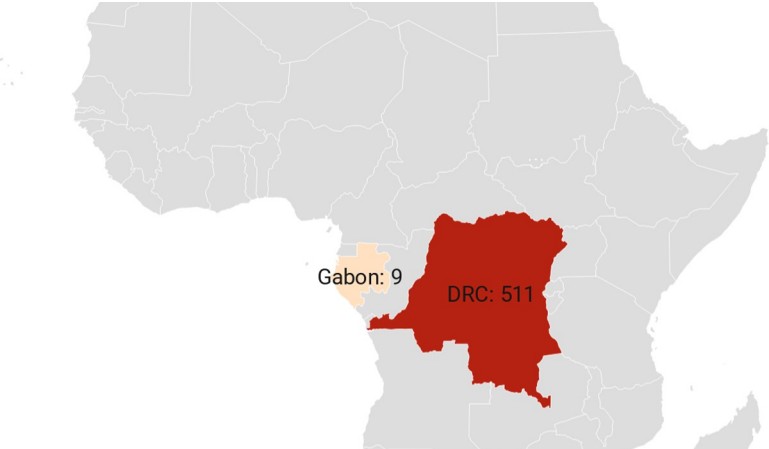

**Fig 4. Number of confirmed, probable, and/or possible monkeypox cases between 1990–1999.** [14,53] (base layer of the map: https://datawrapper.dwcdn.net/EAn8M/1/).

possible cases of monkeypox are the Republic of the Congo (n = 97) and the CAR (n = 69). All other African countries had less than 20 confirmed and probable monkeypox cases each in total over the past five decades.

Monkeypox was not reported outside Africa until 2003, when an outbreak of 47 confirmed or probable cases occurred in the US following exposure to infected pet prairie dogs, which had acquired monkeypox virus from infected exotic animals imported from Ghana [6,40]. In recent years, there have been several travel-associated cases of monkeypox, all following exposures in Nigeria. There was one case in Israel in 2018 [57], three in the UK (two in 2018 [55]; one in 2019 [7]), and one in Singapore in 2019 [8]. A fourth case in the UK (2018) was the result of nosocomial transmission to a healthcare worker [56].

## Number of cases by clade

There are two distinct genetic clades of monkeypox, the Central African (or Congo Basin) clade and the West African clade. Only 10 peer-reviewed articles [8,23,29,35,40,44,48,49,57,58] and one report from the grey literature [63] described specific data on these variants.

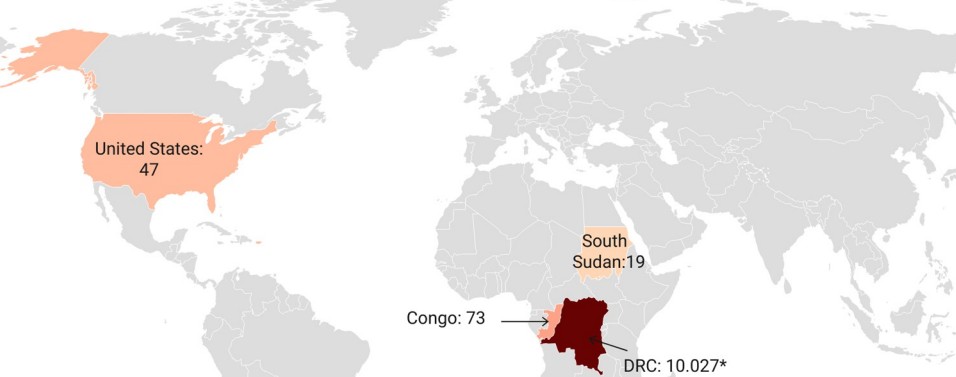

**Fig 5. Number of confirmed, probable, and/or possible monkeypox cases between 2000–2009.** [6,18,46,58,69] * Number reflects suspected cases, since as of the year 2000, the number of suspected cases was primarily reported by the DRC. (base layer of the map: https://datawrapper.dwcdn.net/SXvj7/1/).

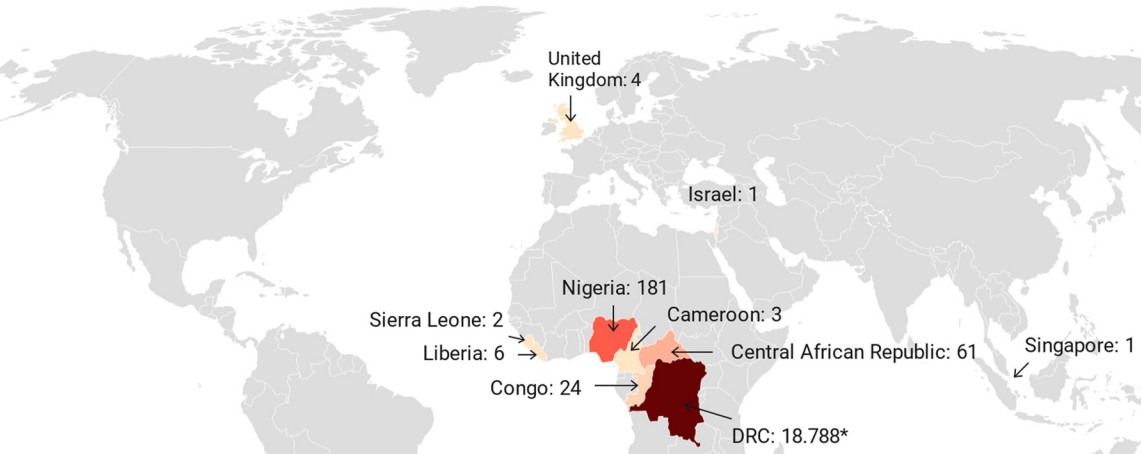

**Fig 6. Number of confirmed, probable, and/or possible monkeypox cases between 2010–2019.** [7,8,15,18,29,30,32,33,35,47–49,55–57,59–67,72–74] * Number reflects suspected cases, since as of the year 2000, the number of suspected cases was primarily reported by the DRC. (base layer of the map: https://datawrapper.dwcdn.net/UUYbg/1/).

Therefore, as noted above in the Methods section, in our calculation of data by clade, we separated the clades according to the geographical division as described by WHO [12].

Table 2 shows the number of cases per clade per decade. A plot of these data (Fig 7) reveals a similar pattern of evolution in the number of cases for both clades. By far, most cases were infected with the Central African clade, which was found in the CAR [29,35], DRC [23,63], and South Sudan [58]. The outbreak in the US (2003) [40] and the outbreak in Nigeria (which started in 2017) [44] cover the largest part of the West African clade cases. This latter clade was also found in Sierra Leone [48,49] and the travel-related cases in Israel [57] and Singapore [8]. Preliminary sequencing data of two UK cases were also determined to be consistent with the West African clade [55].

## Number of suspected cases versus confirmed, probable and/or possible cases

Fifteen peer-reviewed articles and 12 reports from the grey literature described the number of suspected vs. confirmed, probable, and/or possible cases from the various outbreaks (S2 Table). One grey literature source [63] and all but two peer-reviewed articles [15,25] described the number of individuals tested among the suspected cases, and this varied widely from 5% to

**Table 2. Number of Cases per Clade[1].**

| Decade | Central African Clade (N) | West African Clade (N) | Total Cases |
|---|---|---|---|
| 1970–1979 | 38 | 9 | 47 |
| 1980–1989 | 355 | 1 | 356 |
| 1990–1999 | 520 | 0 | 520 |
| 2000–2009 | 92 confirmed<br>10,027 suspected[2] | 47 | 139<br>10,027 |
| 2009–2019 | 85 confirmed<br>18,788 suspected[2] | 195 | 280<br>18,788 |

[1] The five cases from Cameroon are not included in this table, as clade was not reported in any of the articles and WHO reported that Cameroon is the only country in which both clades have been detected [12].

[2] Suspected cases are from the Democratic Republic of the Congo, as number of suspected cases rather than confirmed cases were primarily reported. Suspected cases for other countries are not reported since testing of suspected cases was generally undertaken.

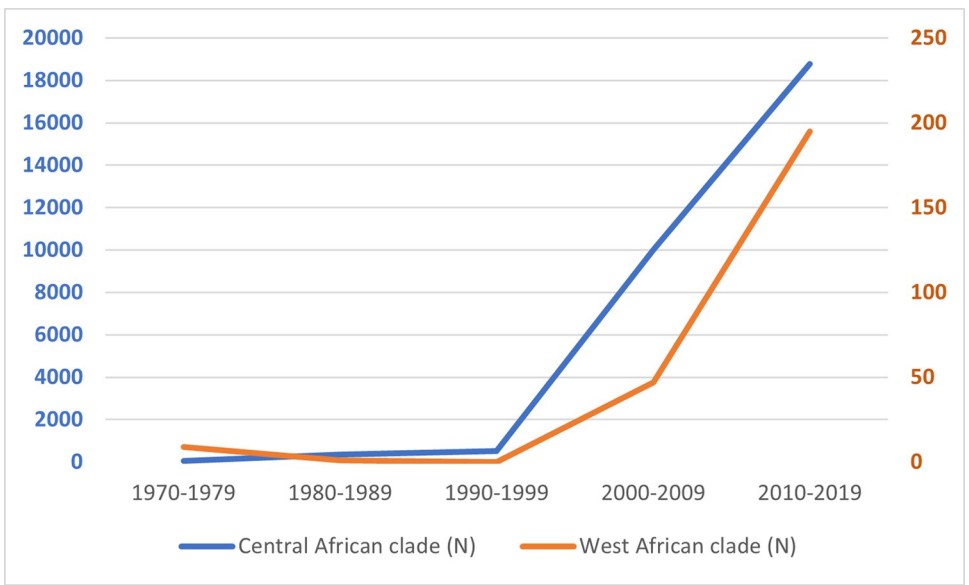

**Fig 7. Evolution of number of cases per clade.** For 2000–2019, the numbers for the Central African clade are based largely on suspected cases, per the reporting system by the Democratic Republic of the Congo.

100%, with the proportion of tested cases found to have confirmed or probable monkeypox ranging from 37.5% to 91.7%. In comparison, none of the 10 WHO reports describe the number suspected cases tested, and in seven of these reports [60,62,65–67,71,73], the percentage of confirmed cases among all tested and non-tested suspected cases was less than 15%.

## Incidence of monkeypox

The incidence rate of monkeypox was reported in only six articles, all peer-reviewed, three with data from the DRC [13,18,28] and three from the CAR [32–34]. Surveillance data of suspected monkeypox cases in the DRC showed that the incidence increased from 0.64/100,000 in 2001 to 2.82/100,000 in 2013 (Fig 8) [18]. Even with the removal of cases from areas of active surveillance, including the Sankuru district of the DRC, the investigators found that the increases remained substantial [18]. Between November 2005 and November 2007, the average annual cumulative incidence of confirmed monkeypox from nine health zones in the Sankuru district was 5.53 per 10,000, ranging from 2.18 to 14.42 per 10,000 [28]. An overall attack rate of confirmed or probable monkeypox in a 2015 outbreak in the CAR was calculated to be 2 per 10,000 persons [32], while an outbreak in 2016 had a reported attack rate of 50 per 10,000 for suspected and confirmed cases [33].

## Secondary attack rate

Only 16 peer-reviewed articles reported secondary attack rates (SARs). Details are presented in S3 Table. A review of these articles did not establish any evolution in SAR over time. More than half of the articles (9/16) reported an SAR of 0% [8,23,30,42,49–52,57], and this spanned the decades from the 1970s through 2010–2019. Similarly, over those same five decades, the SAR ranged from 0.3–10.2% in 6/16 articles [5,14,20,22,54,56]. In the remaining article, a median SAR of 50% was reported in an outbreak among 16 households [25].

## Demographic characteristics

Data on age and sex of confirmed, probable, and/or possible monkeypox cases in Africa are presented in S4 Table. Age was described in 31 peer-reviewed articles and in one report from

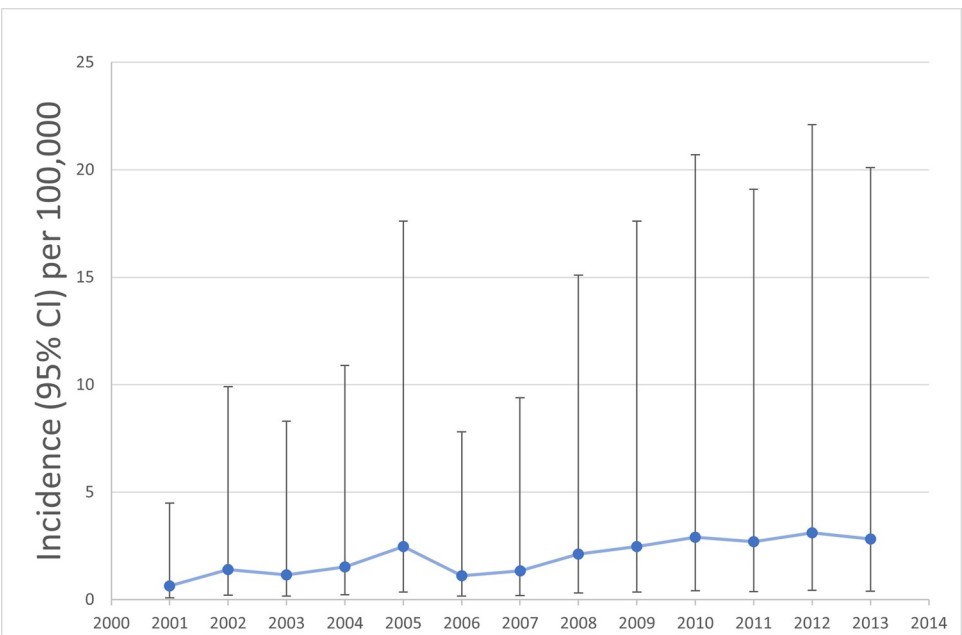

**Fig 8. Incidence rate of suspected monkeypox cases per 100,000 (95% CI) individuals in the DRC, 2001–2013.**
Data from Hoff et al [18].

the grey literature and the sex of individuals was presented in 27 peer-reviewed articles. As shown in Fig 9, the weighted average of the median age of monkeypox infection in Africa has evolved from 4 and 5 years old in the 1970s and 1980s to 10 and 21 years old in the 2000s and

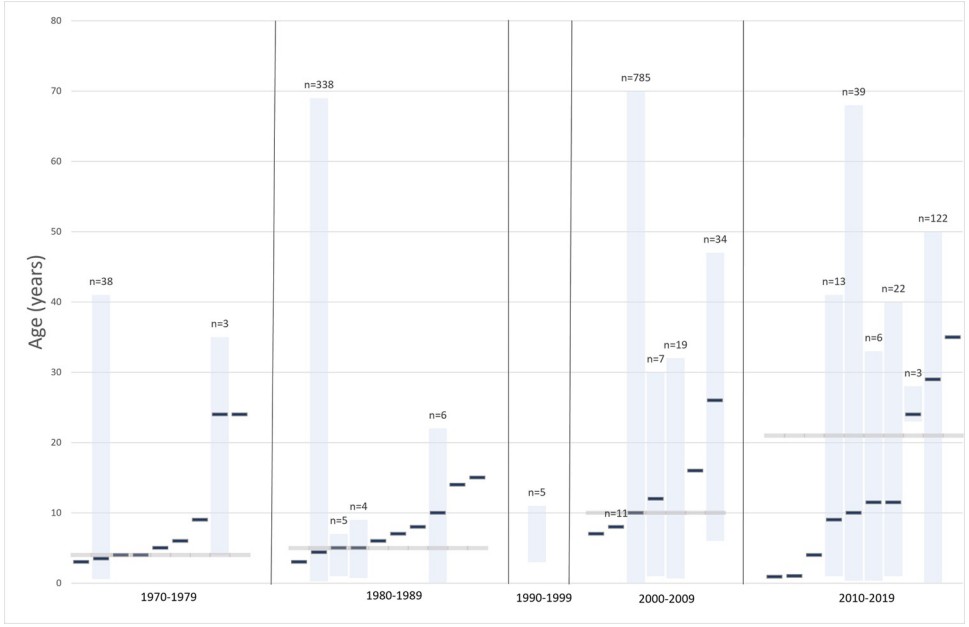

**Fig 9. Median age and range of confirmed, probable and/or possible monkeypox cases in Africa per decade.** Blue bars without range refer to the age of a single case. The grey horizontal line represents the weighted median. No data on median ages could be retrieved for the 1990s.

2010s. Overall, males represented ≥50% of cases in most outbreaks of two or more cases as well as in singular case reports. Cases outside of Africa also occurred more frequently in males and primarily in adults [8,40,55–57]. Only nine peer-reviewed articles [8,15,23,35,44,49,55–57] reported data on occupation in confirmed, probable and/or possible cases. Commonly reported occupations included traders [44], students [44], artisans [44], healthcare professionals [35,44,56], farming [23,44,49], hunting [15], and transportation [35]. Children was listed as an occupation in 8% of 91 cases that reported occupation during the first year of the recent Nigerian outbreak (i.e., Sep 2017 –Sep 2018) [44]. Among the confirmed and further characterized US cases 10 of 34 (29%) were less than 18 years of age [40].

## Smallpox vaccination status

In 21 peer-reviewed articles, information about smallpox vaccination status was reported for confirmed, probable, and/or possible monkeypox cases. In 11 articles, describing outbreaks from 10 different countries, investigators reported that none of the 49 cases were vaccinated. These countries were Cameroon [5,50], Liberia [42], Nigeria [42], Sierra Leone [42], CAR [30,31,34], Republic of the Congo [46], DRC [23], Côte d'Ivoire [51], South Sudan [58], and the UK [56]. The outbreaks in these countries were small, with one to six cases per outbreak, except for the Republic of the Congo with 11 cases [46] and South Sudan with 19 cases [58]. In the 10 other articles, which reported data from outbreaks in the DRC [1981–2013] and US [2003], the proportion of monkeypox cases with a history of prior smallpox vaccination ranged from 4–21% [13,14,19–21,25,26,28,40,41], illustrating that the majority of cases (approximately 80–96%) occurred in unvaccinated individuals. The highest percentage of vaccinated cases (21%) was found in the US outbreak [40]. In a study of confirmed and suspected cases in the CAR, 19.2% (5/26) had a smallpox vaccination scar, and the overall attack rate was lower among vaccinated individuals (0.95/1000) compared to unvaccinated individuals (3.6/1000) [33].

## Case fatality rates

Case fatality rates (CFR) of confirmed, probable, and/or possible cases of human monkeypox were described in 28 peer-reviewed articles and 10 reports from the grey literature, and age at death in 11 peer-reviewed articles and one grey literature report. The details of these data are described in S5 Table. Across all countries, the calculated pooled estimate CFR was 8.7% (Table 3). When the data were separated by clade, the CFR for the Central African clade

**Table 3. Pooled case fatality rate in confirmed, probable, and/or possible monkeypox cases.**

| Countries/Clade | Case Fatality Rate | 95% CI[1] |
|---|---|---|
| All countries[2] | 78/892 = 8.7% | 7.0%– 10.8% |
| Central African clade[3] | 68/640 = 10.6% | 8.4%– 13.3% |
| West African clade[4] | 9/247 = 3.6% | 1.7%– 6.8% |
| West African clade, African countries only | 9/195 = 4.6% | 2.1%– 8.6% |

[1] Exact binomial method (Clopper-Pearson).

[2] The five cases from Cameroon are included in the "all countries" case fatality rate (CFR) calculation, but not in the calculation of CFR by clade, since WHO reported that Cameroon is the only country in which both clades have been detected [12]. The CFR without the inclusion of Cameroon is also 8.7% (77/887).

[3] Central African clade includes the following countries: Central African Republic, Democratic Republic of the Congo, Republic of the Congo, and South Sudan.

[4] West African clade includes the following countries: Côte d'Ivoire, Liberia, Nigeria, Sierra Leone, Israel, Singapore, United Kingdom, and United States.

(10.6%, 95% CI: 8.4–13.3%) was significantly higher than that of the West African clade (3.6%, 95% CI: 1.7–6.8%). When only African countries were included for the West African clade, the trend remained. All nine deaths reported for the West African clade occurred in the recent Nigerian outbreak (which had 181 confirmed or probable cases) [74]. There were no deaths in the cases outside of Africa [6,8,55–57]. Among the reports that included information on age at death, there were a total of 63 deaths (S5 Table). From the 1970s – 1990s, 100% of deaths (47/47) were in children younger than 10 years of age. Over the last two decades (2000–2019), only 37.5% (6/16) of deaths occurred in children <10 years old. A mean age of 27 years was reported for seven deaths among the 122 confirmed or probable cases of monkeypox reported in the first year of the Nigerian outbreak (September 2017-September 2018) [44].

## Mode of transmission and risk factors

In 29 peer-reviewed articles, attempts were made to establish the mode of transmission for confirmed, probable, and/or possible monkeypox cases. The transmission details, by country, including number of cases and mode of transmission, are summarized in S6 Table.

A study from the 1980s involving 338 monkeypox cases from the DRC concluded that an animal source was suspected in 72.5% (245/338) of cases and a human source in 27.5% (93/338) of cases [21]. In contrast, in an investigation of 419 cases from the DRC in the 1990s, only 22% were primary cases (i.e., a person who reported no contact with another person with monkeypox), while 78% were secondary cases (i.e., monkeypox in a person who had contact with an infected person 7–21 days before onset of disease) [14]. Data from the Nigerian outbreak (Sept 2017-Sept 2018) found that transmission was unknown for 62.3% (76/122) of cases [44]. Of the remaining 46 cases, 36 or 78.3% had an epidemiological link to people with similar lesions before the onset of monkeypox and 10 or 8.2% reported contact with animals [44].

All but one of the cases outside of Africa were the result of confirmed or suspected animal-to-human transmission [6,8,41,55,57]. This exception was a human-to-human transmission in the UK in a healthcare worker who provided care to one of the UK confirmed monkeypox cases [56].

The risk factors or risk behaviors for contracting monkeypox were reported in only five studies from three countries (DRC [17,22,26], US [41], Republic of the Congo [45]), and in general reinforced what has been suspected factors. For example, sleeping in the same room or bed, living in the same household, or drinking or eating from the same dish were risk behaviors associated with human-to-human transmission [22,26]. On the other hand, sleeping outside or on the ground or living near or visiting the forest were identified as factors that increase the risk for exposure to animals and subsequent risk for animal-to-human transmission of monkeypox [17,45]. Unexpectedly, assisting with toileting and hygiene and laundering clothes did not have a significant association with acquiring monkeypox, and preparing wild animal for consumption or eating duiker were identified as protective factors [26]. After adjusting for smallpox vaccination status, daily exposure to sick animals (adjusted odds ratio [aOR]: 4.0 (95% CI: 1.2–13.4) or cleaning their cages/bedding (aOR: 5.3 (95% CI: 1.4–20.7) were identified as risk factors for acquiring monkeypox in the 2003 outbreak in the US [41]. Touching or been scratched by an infected animal sufficient to sustain a break in skin were each found to be both significant and nonsignificant risk factors [41].

## Discussion

This systematic review provides a comprehensive overview of the evolution of the epidemiology of monkeypox since it was first detected in humans in 1970. Using a structured format, we describe the greater than 10-fold increase in confirmed, probable, and/or possible monkeypox

cases over the past 5 decades, from 48 cases in the 1970s to 520 cases in the 1990s. Increases in the recent two decades may be confounded by the numbers coming out of the DRC, the country with the most reported cases. Beginning in the year 2000, the DRC started reporting primarily the number of suspected cases, and these have increased from >10,000 cases in 2000–2009 [18] to >18,000 in 2010–2019 [18,63–65]. In the first nine months of 2020 alone, another 4,594 suspected cases were reported in the DRC [66], and the WHO bulletin of the 12-month data for 2020, which was available following completion of this systematic review, reported a total of 6,257 suspected cases [75].

As a result of the recent outbreak, the number of confirmed and probable cases in Nigeria has dramatically escalated as well, from 3 cases in the 1970s [5,42] to 181 cases in 2017–2019 [74]. The surge in cases in the DRC from the 1990s (n = 511) through 2000–2019 (>28,000) is of a similar magnitude. The data from these two countries therefore suggest that the trend is not due to improved reporting alone. This is consistent with the analysis by Hoff and colleagues [18] who found that the rise in monkeypox cases in the DRC from 2001–2013 were likely actual disease increases and not merely a result of improved surveillance, since the reporting system was considered stable by 2008.

There are mounting concerns about the geographical spread and further resurgence of monkeypox. Over the past 5 decades, monkeypox outbreaks have been reported in 10 African countries and 4 countries outside of Africa. In addition to the re-emergence of monkeypox in Nigeria after nearly 40 years, in the years between 2010 and 2019, cases also re-emerged in Liberia and Sierra Leone (after 4 decades) and in the CAR (after 3 decades). First outbreaks emerged in the Republic of the Congo in 2000–2009 and in South Sudan (first appearance in East Africa) in 2005. From 2003, cases of monkeypox have occurred outside of Africa. Infected rodents from Ghana, a country that has not reported any human cases as of this review, were imported into the US. Animal-to-animal transmission then led to animal-to-human transmission, ultimately resulting in an outbreak of 47 confirmed or probable cases [6]. Beginning in 2018 through 2021, adults travelling from Nigeria were diagnosed with monkeypox in Israel [57], the UK [7,55,76], Singapore [8], and the US [77]. These cases were suspected to be the result of animal-to-human transmission. Three additional cases, one resulting from a nosocomial infection and two via transmission to a family member, occurred in the UK [56,76,78]. Of the four monkeypox cases imported into the UK, two have been associated with local transmission and each has resulted in either one or two subsequent cases, illustrating that infected travelers can act as index cases of local outbreaks. Interestingly, the infection imported to the UK in May 2021 [76] and to the US in July 2021[77] occurred at a time where the reported cases of monkeypox in Nigeria were at a very low level. Only 32 suspected cases of disease had been reported to the authorities since the beginning of 2021 [79]. Significant human-to-human transmission has been reported as well in the CAR [30,33,35], DRC [14,21,25], Republic of the Congo [46], South Sudan [58], and Nigeria [5,44], demonstrating the susceptibility of both clades to this type of transmission. Mathematical modelling of human-to-human transmission found that monkeypox has epidemic potential, with $R_0$ >1 [80].

There has been much discussion about the reasons for the resurgence in monkeypox cases, the most prevailing being waning immunity, although deforestation may be a factor or can even act in potentiation [81–83]. Monkeypox virus, variola virus (smallpox), and vaccinia virus (smallpox vaccination) are closely related orthopoxviruses [1]. At the time when smallpox was rampant, no cases of monkeypox were reported. This could have been either because the focus was on smallpox and the presentation of the two diseases are similar or the lack of laboratory confirmation of the etiologic agent led to an assumption of smallpox [84]. Historical data have shown that smallpox vaccination was approximately 85% protective against monkeypox [2]. Following the successful vaccination campaign against smallpox, the disease was

declared eradicated in 1980 by the World Health Assembly, and routine vaccination was halted [3].

Using statistical modeling, Nguyen and colleagues [81] estimated that in 2016, the year before the outbreak in Nigeria began, only 10.1% of the population was vaccinated, and the population immunity, which takes into account waning individual-level immunity, was 2.6%, down from 65.6% in 1970. By 2018, the vaccinated population had decreased to 9.3% and the estimated population immunity had declined to 2.2%. In our review of the literature, we found that unvaccinated individuals accounted for approximately 80–96% of monkeypox cases.

One other possible factor influencing the resurgence of monkeypox might be the genetic evolution of the monkeypox virus. An analysis of the virus genome diversity of 60 samples obtained from humans with primary and secondary cases of infection from Sankuru District, DRC, led to the detection of four distinct lineages within the Central African clade and revealed a gene loss in 17% of the samples that seemed to correlate with human-to-human transmission [85].

Our analysis shows that in the early years (1970–1989), monkeypox was primarily a disease of young children, with a median age at presentation of 4 to 5 years old; this increased to 10 years of age in 2000–2009 and 21 years in 2010–2019. Regarding age at death in monkeypox cases, 100% of deaths were in children <10 years of age in the early years, while for the years 2000–2019, age <10 years accounted for only 37.5% of deaths. These data appear to be consistent with the global intensified smallpox eradication program that began in 1967 [86] and the cease of routine smallpox vaccination by the 1980s following its eradication [3]. In the 2000s, only adults older than 20–25 years would have had a history of smallpox vaccination, leaving the age groups below 20 years vulnerable. Interestingly, the median age of monkeypox cases increased from 10 to 21 years in the next decade. Indeed, most cases were either likely too young to have been vaccinated or were born after the cessation of routine smallpox vaccination, as in the more recent outbreaks.

## Strengths and limitations

The strengths of this review are that basing on Cochrane [10] and PRISMA [11], it included a broad search strategy on monkeypox worldwide, without time or language limits, which reduced selection bias. In addition, there was a thorough review of the grey literature. Overall, more than 60 relevant sources were identified for comprehensive data extraction. There are also limitations.

First, our ability to present a complete picture of the number of confirmed, probable, and/ or possible cases was sometimes limited, as data quantity and quality varied across regions. This is especially true for Central African countries, and in particular the DRC, where a systematic accounting and reporting of the number of cases per year is lacking. The number of cases presented in the maps in this review, especially after 1986 when WHO stopped their surveillance program in the DRC, are likely lower than the actual number of cases [28] and underreporting is quite probable, despite implementation of the Integrated Disease Surveillance and Response in the DRC in 2000 [18]. Furthermore, no national estimates of the number of confirmed cases in the DRC can be made, as polymerase chain reaction testing is infrequently performed in the field [18,25]. Second, there is a paucity of data on the age of cases, which could call into question the results of our analysis of median age at monkeypox diagnosis. For example, a report from the DRC, published after completion of this systematic review, found a median age of 14 years for confirmed cases in the Tshuapa Province during 2011–2015 [87]. These investigators did note, however, that median age at onset has increased over time [87], which corresponds to our findings. Furthermore, our review of the literature found that age at

death from monkeypox has also increased, which is then consistent with our age-at-diagnosis findings. Third, since specific data on clades was infrequently reported, we assigned clades based upon the geographical spread described by WHO [12] and drew conclusions about both number of cases per clade and mortality per clade. These results did not allow for the possibility that one clade may invade other geographies or consider that factors other than clade (e.g., access to healthcare) might account for the mortality differences. Fourth, although more than half of the included articles presented data on transmission of monkeypox, in many studies not all cases could be definitively attributed to either animal-to-human transmission or human-to-human transmission. Therefore, an in-depth analysis about the proportion of cases being infected by human-to-human transmission could not be performed. Fifth, while a changing epidemiology of monkeypox could be linked to the genetic evolution of the monkeypox virus, a review of the literature on the latter was not within scope of our work. Lastly, data on risk factors for acquiring monkeypox are rather scarce, and some incongruous results were found. In one study, for example, eating duiker and preparing wild animal for food were identified as protective factors [26], which appears in contrast to factors identified in animal-to-human transmission cases [15,29]. Thus, more formal studies of risk factors are warranted.

## Conclusions

The waning population immunity associated with discontinuation of smallpox vaccination has established the landscape for the resurgence of monkeypox. This is demonstrated by the increases in number of cases and median age of individuals acquiring monkeypox as well as the re-emergence of outbreaks in some countries after an absence of 30–40 years. Further, the appearance of cases outside of Africa highlights the risk for geographical spread and the global relevance of the disease. The possibility for human-to-human transmission is a concern not just among household members, but also among providers of care to diseased individuals. In light of the current environment for pandemic threats, the public health importance of monkeypox disease should not be underestimated. International support for increased surveillance and detection of monkeypox cases are essential tools for understanding the continuously changing epidemiology of this resurging disease.

## Supporting information

**S1 Table. Number of monkeypox cases by decade by country.**
(DOCX)

**S2 Table. Number of suspected cases versus confirmed, probable, and/or possible cases.**
(DOCX)

**S3 Table. Secondary attack rate.**
(DOCX)

**S4 Table. Age and sex of confirmed, probable, and/or possible and hospitalised cases from Africa.**
(DOCX)

**S5 Table. Case fatality rate in confirmed, probable, and/or possible monkeypox cases.**
(DOCX)

**S6 Table. Transmission of monkeypox.**
(DOCX)

## Acknowledgments

The authors would like to thank Babette van Deursen (BVD), Rosa van Hoorn (RVH), Lauren Mason (LM), and Femke van Kessel (FVK), current (LM and FVK) and former (BVD and RVH) employees of Pallas Health Research and Consultancy, for assistance with performing the literature searches (BVD, LM), quality review of the findings (RVH), and creating figures and maps (FVK).

## Author Contributions

**Conceptualization:** Bernard Hoet.

**Data curation:** Bernard Hoet.

**Formal analysis:** Liddy Chen.

**Funding acquisition:** Bernard Hoet.

**Investigation:** Eveline M. Bunge, Florian Lienert, Heinz Weidenthaler.

**Methodology:** Eveline M. Bunge, Bernard Hoet, Liddy Chen, Heinz Weidenthaler.

**Project administration:** Bernard Hoet, Florian Lienert.

**Resources:** Bernard Hoet.

**Supervision:** Bernard Hoet, Robert Steffen.

**Validation:** Bernard Hoet, Liddy Chen, Florian Lienert, Heinz Weidenthaler, Robert Steffen.

**Visualization:** Bernard Hoet, Liddy Chen, Robert Steffen.

**Writing – original draft:** Lorraine R. Baer.

**Writing – review & editing:** Eveline M. Bunge, Bernard Hoet, Liddy Chen, Florian Lienert, Heinz Weidenthaler, Lorraine R. Baer, Robert Steffen.

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
