## [Decision Letter · Decision Letter 0]

19 Oct 2021

Dear MD Hoet,

Thank you very much for submitting your manuscript "The changing epidemiology of human monkeypox – a potential threat? A systematic review" for consideration at PLOS Neglected Tropical Diseases. As with all papers reviewed by the journal, your manuscript was reviewed by members of the editorial board and by several independent reviewers. The reviewers appreciated the attention to an important topic. Based on the reviews, we are likely to accept this manuscript for publication, providing that you modify the manuscript according to the review recommendations. 

Sincerely,

Gregory Gromowski

Associate Editor

A. Desiree LaBeaud

Deputy Editor

Reviewer's Responses to Questions

**Key Review Criteria Required for Acceptance?**

**Methods**

-Are the objectives of the study clearly articulated with a clear testable hypothesis stated?

-Is the study design appropriate to address the stated objectives?

-Is the population clearly described and appropriate for the hypothesis being tested?

-Is the sample size sufficient to ensure adequate power to address the hypothesis being tested?

-Were correct statistical analysis used to support conclusions?

-Are there concerns about ethical or regulatory requirements being met?

Reviewer #1: It was a well designed systematic review of the literature on monkeypox that was registered with PROSPERO

Reviewer #2: The manuscript presents a systematic review of monkeypox epidemiology with a focus on changes in the evolution since the first cases in the 1970s through the time of research. The review followed the PRISMA guideline and was prospectively registered with PROSTERO. The methods are well-explained. The study design, defined population (Africa and Ex-Africa) and analysis are appropriate for the study objective. 

Competing interest by the funder and initiator of the study, Bavarian Nordic ( manufacturers of monkeypox vaccines) and the authors are declared.

**Results**

-Does the analysis presented match the analysis plan?

-Are the results clearly and completely presented?

-Are the figures (Tables, Images) of sufficient quality for clarity?

Reviewer #1: The analysis was thorough, informative and well presented. The information presented is timely.

Reviewer #2: The analysis is descriptive and matches the analysis plan. However, there are some missing components that are critical. 

Given the focus on describing the epidemiology of monkeypox, one gap was an absence of an analysis of genomics of the virus. An important recommendation for the authors to consider is an appraisal of the genetic evolution of the virus. Some studies to consider include: https://doi.org/10.1093/infdis/jiaa559 ; https://doi.org/10.3201/eid2002.130118, https://doi.org/10.1016/S1473-3099(18)30043-4

**Conclusions**

-Are the conclusions supported by the data presented?

-Are the limitations of analysis clearly described?

-Do the authors discuss how these data can be helpful to advance our understanding of the topic under study?

-Is public health relevance addressed?

Reviewer #1: Conclusions were well supported by the results cited. Resurgence of monkeypox is an important public health issue.

Reviewer #2: The conclusion emphasised the available information and the need for improved surveillance to better understand the epidemiology of the disease

**Editorial and Data Presentation Modifications?**

Reviewer #1: One minor issue concerns the authors statement on lines 483-484 that unvaccinated individuals accounted for approximately 80-96% of monkeypox cases. The authors should explain and document in more detail how this percentage range was determined.

Reviewer #2: Figure 2,3,4,5,6 would be better presented in a table for better readability

Line 318 - article should be articles

Line 419-421 - The statement " Conflicting findings of significance, however, were reported regarding direct exposure or touching the animal or having received a bite or scratch sufficient to break the skin" is not clear. Please, state the conflict being referred to. Though this is mentioned in the discussion, it needs to be stated in the results.

References 38 and 39 are same - please, edit.

**Summary and General Comments**

Reviewer #1: This was a well-crafted, comprehensive review of the epidemiology of monkeypox which constitutes a valuable contribution to the field.

Reviewer #2: This is a relevant study which summarises available evidence of monkeypox epidemiology at a time of increasing awareness of ongoing transmission in Africa and exportation of cases with need for development of control strategies. Though a similar review was published in 2018 (cited), this study marginally more informational by including newly published works. Studies with phylogenetic analysis are excluded. Phylogenetic analysis is important to describe the evolution of monkeypox in the past 4-5 decades.

PLOS authors have the option to publish the peer review history of their article (what does this mean?). If published, this will include your full peer review and any attached files.

Reviewer #1: No

Reviewer #2: No

Figure Files:

Data Requirements:

Reproducibility:

References

---

## [Editor Report · Decision Letter 1]

4 Jan 2022

Dear MD Hoet,

We are pleased to inform you that your manuscript 'The changing epidemiology of human monkeypox – a potential threat? A systematic review' has been provisionally accepted for publication in PLOS Neglected Tropical Diseases.

Best regards,

Gregory Gromowski

Associate Editor

A. Desiree LaBeaud

Deputy Editor

---

## [Editor Report · Acceptance letter]

7 Feb 2022

Dear MD Hoet,

We are delighted to inform you that your manuscript, "The changing epidemiology of human monkeypox – a potential threat? A systematic review," has been formally accepted for publication in PLOS Neglected Tropical Diseases.

Best regards,

Shaden Kamhawi

co-Editor-in-Chief

Paul Brindley

co-Editor-in-Chief
